# HMR-Adapter: A Lightweight Adapter with Dual-Path Cross Augmentation for Expressive Human Mesh Recovery

## ABSTRACT

Expressive Human Mesh Recovery (HMR) involves reconstructing the 3D human body, including hands and face, from RGB images. It is difficult because humans are highly deformable, and hands are small and frequently occluded. Recent approaches have attempted to mitigate these issues using large datasets and models, but these solutions remain imperfect. Specifically, whole-body estimation models often inaccurately estimate hand poses, while hand expert models struggle with severe occlusions. To overcome these limitations, we introduce a dual-path cross augmentation framework with a novel adaptation approach called HMR-Adapter that enhances the decoding module of large HMR models. HMR-Adapter significantly improves expressive HMR performance by injecting additional guidance from other body parts. This approach refines hand pose predictions by incorporating body pose information and uses additional hand features to enhance body pose estimation in whole-body models. Remarkably, a HMR-Adapter with only about 27M parameters achieves better performance in fine-tuning the large model on a target dataset. Furthermore, HMR-Adapter significantly improves expressive HMR results by combining the adapted large whole-body and hand expert models. We show extensive experiments and analysis to demonstrate the efficacy of our method.

## CCS CONCEPTS

• **Computing methodologies** → **Reconstruction**.

## KEYWORDS

expressive human mesh recovery, adapter, transformer

## 1 INTRODUCTION

Expressive Human Mesh Recovery (HMR) is a long-standing problem in computer vision, aiming to estimate the whole-body 3D human shape and pose, facial expressions, and hand gestures from an RGB image. This is a crucial task as it underpins various downstream applications, e.g., clothed human reconstruction, virtual try-on, and AR / VR content generation. However, this task is quite challenging because humans are highly deformable, and there is usually blur or occlusion with their movement. Moreover, capturing the motion of hands is extremely hard since the image sizes

of hands and faces are relatively small while exhibiting complex articulations.

To address the aforementioned problems, existing research on expressive HMR generally falls into two categories: multiple-stage methods [4, 20, 39] and single-stage methods [3, 22, 24]. Multiple-stage methods reconstruct the body [11, 17, 21, 43], hands [2, 13, 14, 29], and faces [5, 34, 35, 37, 46] with corresponding expert models individually, then different parts can be directly combined to achieve whole-body reconstruction due to the consistency of the whole-body parametric model, such as SMPL-X [28]. However, such a straightforward strategy may lead to unnatural integration due to the lack of holistic modeling. On the contrary, one-stage methods reconstruct the whole-body posture, face, and hand pose in a unified manner. However, minor rotation errors are accumulated along the kinematic chain, leading to noticeable drifts in distal joint positions, e.g., hand joints. Moreover, the hand joints suffer from small input resolutions, making it challenging to capture the variations of complex hand gestures. Both paradigms have not effectively addressed the collective modeling of hands and other body parts. Instead, we opt for a new framework that leverages cross-body-part cues to improve both body and hand modeling.

Recent works [3, 11, 29] scale up the model size and training data to fully unlock the potential of massive human motion datasets. These methods adopt large-scale image encoders as their backbone to extract visual cues in the input image, followed by several independent decoder modules used to predict the human model parameters. However, these large HMR models remain limited. Specifically, SMPLer-X [3], a whole-body mesh recovery model, inaccurately estimates hand poses by cropping small hand feature maps. It also fails to account for interactions between body parts, resulting in noticeable errors in distal joint positions, e.g., hand joints, because of the accumulated drifts along the kinematic chain. HaMeR [29] is a large hand expert model to process hand images exclusively. However, human movements are holistic and interconnected, with hand actions being significantly influenced by the body. HaMeR's limitation lies in its focus on only the hand area, lacking global semantic context. This results in performance degradation in scenarios where hands are severely occluded or blurred.

To address these problems above, we introduce a promising, lightweight framework, named HMR-Adapter, with dual-path cross augmentation. We adapt the existing large-scale HMR models and further boost their performance for expressive HMR.

First, we use the dual-path cross augmentation paradigm to enhance both the body pose and hand pose estimation. We observe that different body parts are inherently interconnected: Body poses significantly affect the global orientation of the hands in various ways, while the positions of hand joints can indicate the potential range of arm poses. We first adapt a hand expert model to strengthen its robustness under occlusion and blur cases. Specifically, we introduce the body pose as additional guidance for more precise hand

(a) Predict hand pose with a large hand expert model

(b) Large hand expert model with our HMR-Adapter

**Figure 1: Comparison between using an existing large-scale hand expert model to predict hand poses and ours. The hand expert can only see the hand image. The result (from [29]) produces inaccurate hand prediction when the hand is severely occluded. In contrast, we adapt the large hand expert model and inject additional body pose guidance to help the hand expert understand the overall context, thus giving reasonable predictions under occlusion.**

pose estimation. In the meantime, we adapt a whole-body HMR model to refine the upper body pose estimation and encourage natural integration between the hand and body. Specifically, we use hand information to further refine the initial predicted body pose from the whole-body model's body pose regressor. To this end, we sample image features from the finger's root joints (MCP joints) as additional guidance to predict the body pose offset. This will not only enhance the body pose but also smooth out the poses at intersecting joints between different body parts (e.g., wrists).

Second, we propose HMR-Adapter to adapt the target model while maintaining the strong capability of pre-trained expert models. In contrast to previous methods [8, 24, 45] that combine body feature and hand feature by concatenating or weighted sum, HMR-Adapter does not harm the original model's structure and knowledge. The adapter is parallel with the original hand expert's decoder, combining hand features and body context without modifying the original decoder. While the previous method [24] claimed that body feature contains much unnecessary information which may harm the hand pose estimation, we find that with our HMR-Adapter, critical information such as the motion of the wrists and arms are getting much greater attention when predicting hand poses. We argue that the main reason why directly adding body information harms hand estimation lies in the naive incorporation of body information, which can potentially confuse the model by obscuring critical information. In our proposed HMR-Adapter, body information from different parts is adaptively selected to aid hand prediction. In this way, the adapted hand expert model is more robust and can leverage useful body information to perform better in occlusion and truncation situations.

Another interesting property of our HMR-Adapter design is that it is lightweight and very efficient. When fine-tuning the large HMR models, the performance of a HMR-Adapter with around 30M parameters is comparable to or even better than a fully fine-tuned large model. With the proposed HMR-Adapter, the large HMR models can converge faster, even without additional guidance injection.

Quantitative and qualitative experimental results show that our HMR-Adapter is efficient and generalizable, which can be applied to large HMR models with different architectures. A small HMR-Adapter with about 30M parameters is comparable to or even better than the fully fine-tuned models for HMR fine-tuning. Besides,

HMR-Adapter significantly improves the large whole-body model in terms of hand estimation. Our proposed dual-path cross augmentation with HMR-Adapter for expressive HMR significantly outperforms previous approaches on the challenging ARCTIC dataset, where severe occlusions occur.

Our contributions can be summarized as follows.

- We present a promising lightweight but efficient adaptation method called HMR-Adapter that can be applied across large-scale HMR models. A small HMR-Adapter with around 30M parameters is comparable to or even better than fine-tuning the entire large model.
- We propose a dual-path cross augmentation paradigm to jointly optimize the body pose and hand pose estimation in expressive HMR. This contributes to more robust hand reconstruction and more accurate body pose prediction. We highlight this on ARCTIC, a hand-object interaction dataset with heavy occlusion.
- We demonstrate that our framework for expressive HMR outperforms several state-of-the-art methods for whole-body mesh reconstruction.

## 2 RELATED WORK

### 2.1 Expressive Human Mesh Recovery

Expressive human mesh recovery focuses on reconstructing human meshes, including body, hands, and face, from a single image. Early research employed predefined parametric models like SMPL-X [28] for prediction. Current approaches to expressive human mesh recovery can be broadly divided into two groups: multi-stage and single-stage methods. Multi-stage approaches break down the reconstruction problem into parts, addressing the estimation of the human body [11, 17, 19, 40], hands [2, 13, 14, 29], and faces [5, 34, 35, 37, 46] separately using part-specific models, before combining with a body template layer. This strategy may produce artifacts at intersecting joints.

In contrast, one-stage paradigm proposes to reconstruct all body components as a whole. ExPose [4] predicts hands, face, and body SMPL-X parameters, using body estimation to refine face and hand regions from high-resolution inputs. PIXIE [8] calculates part-specific feature confidence and merges face-body and hand-body features for robust regression, while Hand4Whole [24] identifies

joint-level features essential for body and hand joint rotations. PyMAF-X [39] and HybrIK-X [20] propose smooth integration methods between different body parts to integrate results from sub-networks. OSX [22] uses a transformer-based end-to-end method for body part connections. RoboSMPLX [27] improves body part localization and feature extraction for robust whole-body model recovery. SMPLer-X [3] scales up models and data for whole-body recovery. Despite recent advancements in one-stage methods, effectively recovering whole-body models with realistic hand gestures remains challenging, particularly in scenarios involving occlusions, interacting hands, and motion blur. To this end, We adapt existing large-scale HMR models, incorporating a dual-path augmentation strategy between body parts and hands to enhance their performance.

## 2.2 Large Model Adaptation

Standard full fine-tuning of large models is computationally intensive and can degrade generalization. A prevalent solution is to use adapters to keep the pre-trained backbone intact, incorporating only a minimal number of strategically positioned trainable parameters within the architecture. Adapters have been extensively utilized in the field of large language models [10, 16, 26, 33, 41, 42] and are increasingly applied in visual-language alignment models. CLIP-Adapter [9] integrates residual-style adapters after text and visual encoders of CLIP [30] to avoid the gradients back-propagation through CLIP's encoders. IP-Adapter [38] introduces a novel decoupled cross-attention mechanism that differentiates between text and image features, adding a trainable cross-attention layer for image features within each existing layer.

Despite the advancement in large model adaptation, the area of adapting large human mesh recovery models remains rarely explored. We propose our light-weight HMR-Adapter to incorporate additional guidance information while retaining the extensive knowledge in the pre-trained HMR models.

## 3 METHOD

### 3.1 Overview

The pipeline of using our proposed HMR-Adapter for expressive human motion capture is presented in Fig. 2. Given a human image, our task is to predict the pose and shape parameters of the SMPL-X [28] model. We adopt a dual-path pipeline to approach this: In one path, a pre-trained whole-body model predicts the initial body pose and the hand bounding boxes. These bounding boxes are then used to crop out the hand area from the input image. With the sampled hand image features, HMR-Adapter refines the initial body pose. In the other path, HMR-Adapter augments hand pose prediction from a pre-trained hand expert model by injecting body pose information. Finally, the refined body pose and hand pose are forwarded to the SMPL-X layer to get the recovered human mesh.

### 3.2 Preliminary

#### 3.2.1 Parametric Human Models.
Our task is to recover the whole-body human mesh in a parameterized way, which is to predict the pose and shape parameters of the SMPL-X human model [28], a unified model that models the human body, hands, and face. SMPL-X takes as input the pose parameters $\theta \in \mathbb{R}^{55 \times 3}$ that includes body,

hands, and jaw poses; shape parameters $\beta \in \mathbb{R}^{10}$, and expression parameters $\psi \in \mathbb{R}^{10}$. SMPL-X defines a function $M(\beta, \theta, \psi)$ that returns the mesh of the human body $V \in \mathbb{R}^{(N \times 3)}$, with $N = 10475$ vertices. SMPL-X additionally returns the $J = 53$ major joints, which include 22 main body joints, 1 jaw joint, and 15 joints per hand.

We use the MANO parametric hand model [32] for hand modeling. MANO shares the same pose space with SMPL-X's hand part. MANO also defines a function that takes hand pose parameters $\theta^h \in \mathbb{R}^{16 \times 3}$ that includes 3D hand global rotation and 3D finger rotations, and hand shape parameters $\beta^h \in \mathbb{R}^{10}$. MANO returns the mesh of the hand $V^h \in \mathbb{R}^{(N' \times 3)}$, with $N' = 778$ vertices.

#### 3.2.2 Hand Expert Model.
To better understand the use of HMR-Adapter, we first briefly introduce the original hand expert model. The hand model adopts a fully transformer-based design. Before entering the backbone, the hand image is resized to $I_h$ with a shape of $256 \times 256$. The image is then split into fixed-size patches with patch embedding to obtain the hand image tokens. The tokens are further processed with a ViT [6] backbone that follows the "huge" design (denoted as ViT-Huge). The backbone further processes the image tokens to get a series of output tokens $\mathbf{F}^H$, followed by a transformer head that regresses the hand parameters. The transformer head has multiple transformer blocks. Each block consists of a multi-head self-attention, a cross-attention, and a feed-forward network (FFN). The transformer head receives a single query token while cross-attending to the hand image tokens $\mathbf{F}^H$. Specifically, the single query token $z$ is first mapped to a higher dimension using a FFN, then encoded by a self-attention module to form the cross-attention query features $\mathbf{Z}$. Formally, the decoder incorporates the query features $\mathbf{Z}$ and the input image feature $\mathbf{F}^H$ in a cross-attention manner:

$$\mathbf{Z}' = \text{CA}(\mathbf{Q}, \mathbf{K}, \mathbf{V}) = \text{Softmax}\left(\frac{\mathbf{Q}\mathbf{K}^\top}{\sqrt{d}}\right)\mathbf{V}, \tag{1}$$

where $\mathbf{Q} = \text{SA}(\mathbf{Z})\mathbf{W}_q, \mathbf{K} = \mathbf{F}^H\mathbf{W}_k, \mathbf{V} = \mathbf{F}^H\mathbf{W}_v$ are the query, key, and values matrices of the cross-attention layer respectively, and $\mathbf{W}_q, \mathbf{W}_k, \mathbf{W}_v$ are the weight of the trainable linear projection layers. The output of the head $\mathbf{Z}^{\text{new}}$ is passed to three independent linear layers to predict the hand pose parameters, camera parameters, and hand shape parameters, respectively.

#### 3.2.3 Whole-Body Model.
Similarly, the whole-body model preprocesses the image features with a ViT, which additionally learns a series of task tokens concatenated with the image tokens. Hand and face bounding boxes are predicted from the image tokens. The hand and body heads that predict body pose consist of a positional module to predict 3D keypoints and a regressor module to predict pose parameters. And we concentrate on adapting the body regressor module. The original body regressor predicts 3D body joint rotations $\theta_0$ (including the root joint rotation) by a linear layer, using a concatenation of 3D body joint coordinates and task tokens as input.

## 3.3 Hand Expert Model Adaptation

Since we are putting our hand expert model in a whole-body HMR setting, it is natural to introduce information from the human body into hand pose prediction. Such body guidance information enables

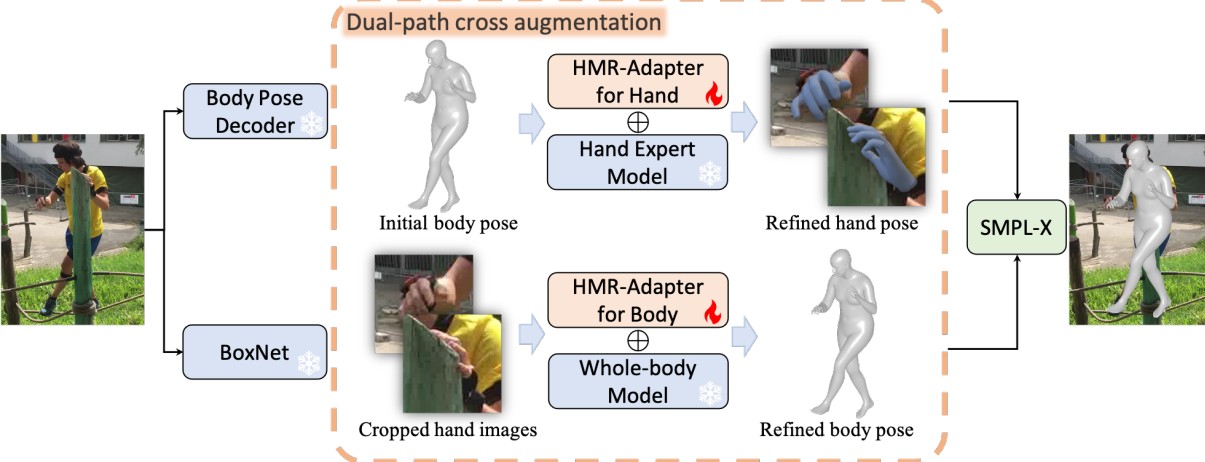

Figure 2: The overview of our HMR-Adapter with dual-path cross augmentation for expressive HMR. HMR-Adapter refines body pose with additional hand features in one path and enhances hand pose estimation by introducing extra body guidance in the other path. The refined body pose and hand pose are combined together to the SMPL-X layer to get the final human mesh. Only the two HMR-Adapters (in orange color) can be trained during training.

the hand model to predict reasonable hand pose even under severe occlusion.

We use body pose as the guidance instead of the commonly used body image feature since we do not have access to the whole-body image when training HMR-Adapter on the hand expert. To prevent unrelated information (such as the motion of lower limbs) from corrupting hand pose estimation, we choose joints from the root joint to the Left / Right Wrist along the kinematic chain, named the Left / Right Arm joint group. Each joint's rotation is represented in the 6D rotational representation of [44]. The Arm joint group for each hand is denoted as $\theta_b \in \mathbb{R}^{J \times 6}$, where $J$ is the number of arm joints. Then, we apply a linear layer to map $\theta_b$ to produce the body joint conditioning features $\mathbf{F}^B \in \mathbb{R}^{J \times 1280}$ aligned with the hand image feature. We set the additional body pose to zero when training on hand-only datasets that do not contain body pose labels.

A straightforward method for incorporating the body pose feature involves concatenating it with the original hand image feature $\mathbf{F}^H$ and subsequently inputting them into the decoder head. Alternatively, following the method described in [8], a moderator could be used to merge the body pose and image feature through a weighted sum. However, these simple techniques of concatenation or summation could improperly couple these features, failing to fully leverage the injected body pose feature and potentially causing the "catastrophic forgetting" [31] of the pre-trained hand expert model. In practice, we have observed that such naive approaches can deteriorate the model's performance.

To this end, we propose to use a decoupled transformer module to incorporate the body pose guidance, with separate transformer blocks for the body pose feature and hand image feature. Specifically, we add a new self-attention and cross-attention layer for each transformer decoder block in the original hand decoder head to insert the body pose feature. Given the body pose feature $\mathbf{F}^B$, the

output of new cross-attention $\mathbf{Z}''$ is computed as follows:

$$\mathbf{Z}'' = \text{CA}\left(\mathbf{Q}', \mathbf{K}', \mathbf{V}'\right) = \text{Softmax}\left(\frac{\mathbf{Q}'\,(\mathbf{K}')^\top}{\sqrt{d}}\right)\mathbf{V}', \quad (2)$$

where $\mathbf{Q}' = \text{SA}'(\mathbf{Z})\mathbf{W}'_q$, $\mathbf{K}' = \mathbf{F}^B\mathbf{W}'_k$ and $\mathbf{V}' = \mathbf{F}^B\mathbf{W}'_v$ are the query, key, and value matrices from the body pose feature. $\mathbf{W}'_k$ and $\mathbf{W}'_v$ are the corresponding weight matrices. SA$'$ is the new self-attention layer. In order to speed up the convergence, SA$'$, $\mathbf{W}'_q$, $\mathbf{W}'_k$ and $\mathbf{W}'_v$ are initialized from SA, $\mathbf{W}_q$, $\mathbf{W}_k$ and $\mathbf{W}_v$. Then, we add the output of body pose cross-attention to the output of hand image cross-attention. Hence, the final formulation of the decoupled transformer layer is defined as follows:

$$\begin{aligned}\mathbf{Z}^{\text{new}} &= \text{CA}(\mathbf{Q}, \mathbf{K}, \mathbf{V}) + \text{CA}\left(\mathbf{Q}', \mathbf{K}', \mathbf{V}'\right)\\&\text{where } \mathbf{Q} = \text{SA}(\mathbf{Z})\mathbf{W}_q, \mathbf{K} = \mathbf{F}^H\mathbf{W}_k, \mathbf{V} = \mathbf{F}^H\mathbf{W}_v, \quad (3)\\&\qquad \mathbf{Q}' = \text{SA}'(\mathbf{Z})\mathbf{W}'_q, \mathbf{K}' = \mathbf{F}^B\mathbf{W}'_k, \mathbf{V}' = \mathbf{F}^B\mathbf{W}'_v\end{aligned}$$

Since we freeze the original hand decoder, only the $\mathbf{W}'_q, \mathbf{W}'_k, \mathbf{W}'_v$ and SA$'$ are trainable in the above motion-adapter. Finally, the output token $\mathbf{Z}^{\text{new}}$ is passed to the original linear layer for predicting hand poses.

## 3.4 Whole-body Model Adaptation

We also augment body pose prediction using information from hands. Inspired by [24], we use the hand image feature sampled by predicted hand joints to benefit body pose prediction. We extract hand joint features from the 2D hand feature map. Specifically, we use the fingers' root joints (i.e., hand MCP joints) from both hands, and the 2D hand feature map is obtained by using the ROI (regions of interest) module to crop features from the whole-body image's feature map with the predicted hand bounding boxes. A 1-by-1 convolution reduces the 2D map's channel dimension from 1280 (ViT-Huge feature dimension) to 512, followed by bilinear

interpolation at the MCP joint coordinates, yielding joint features $\mathbf{F}^J \in \mathbb{R}^{8 \times 512}$ for the four MCP joints per hand.

Similar to adapting the hand expert, we use parallel transformer blocks to incorporate the additional hand joint feature, with the original body regressor fixed. The new transformer layer receives a single query token as an input token and predicts the body pose offset $\Delta\theta$ to the initial body pose prediction $\theta_0$. Formally, the refined body pose $\Delta\theta$ can be defined as:

$$\hat{\theta} = \theta_0 + \Delta\theta = \theta_0 + \text{FFN}\left(\text{CA}(\mathbf{Q}'', \mathbf{K}'', \mathbf{V}'')\right), \quad (4)$$

where FFN is the body pose offset regressor, $\mathbf{Q}'' = \text{SA}''(\mathbf{Z}'')\mathbf{W}_q''$, $\mathbf{K}'' = \mathbf{F}^J\mathbf{W}_k''$, $\mathbf{V}'' = \mathbf{F}^J\mathbf{W}_v''$ are the query, key, and value matrices of the attention operation respectively, and $\mathbf{W}_q'', \mathbf{W}_k'', \mathbf{W}_v''$ are the weights of the trainable linear projection layers. SA$''$ is the trainable self-attention layer. The query token $\mathbf{Z}''$ is an empty beginning token.

## 3.5 Training

We adopt a two-stage training style to train the two HMR-Adapters separately. During training, we freeze all the original modules in the hand expert model and the whole-body model and only train the proposed two HMR-Adapters.

*3.5.1 Hand Model.* We use the ground truth body pose as the conditioning body pose input when training HMR-Adapter for the hand expert model. We aim to optimize the weights of HMR-Adapter to introduce body pose guidance properly. The objective function we aim to minimize is defined as:

$$\mathcal{L} = \lambda_{2D}\mathcal{L}_{2D} + \lambda_{3D}\mathcal{L}_{3D} + \lambda_v\mathcal{L}_v + \lambda_\theta\mathcal{L}_\theta + \lambda_\beta\mathcal{L}_\beta \quad (5)$$

where $\lambda_{2D/3D}$ stands for the loss weights for the 2D/3D joints error, $\lambda_v$ denotes the loss weight of vertex error, and $\lambda_{\theta/\beta}$ is for the MANO hand pose and shape parameters $\theta/\beta$. The hand pose parameter loss $\mathcal{L}_\theta$ is a combination of L1 and cosine distance between the predicted hand pose $\theta$ and the target pose $\theta^*$:

$$\mathcal{L}_\theta = \left\| \theta - \theta^* \right\|_2^2 + (1 - \cos(\theta, \theta^*)), \quad (6)$$

where cos indicates the cosine similarity between two items. Note that we do not use hand pose adversarial loss, which reduces the training cost.

*3.5.2 Body Model.* When training HMR-Adapter for the whole-body large model, we use the ground truth 2D hand keypoints to sample from the hand image feature map and use the sampled joint features as the body HMR-Adapter's additional guidance. We optimize the weights of HMR-Adapter to encourage integrating crucial hand information into the whole-body model's body prediction head. We use the following loss function when adapting the whole-body model:

$$\mathcal{L} = \lambda_{2D}\mathcal{L}_{2D} + \lambda_{3D}\mathcal{L}_{3D} + \lambda_\theta\mathcal{L}_\theta + \lambda_\beta\mathcal{L}_\beta \quad (7)$$

where $\lambda_{2D/3D}$ stands for the loss weights for the 2D/3D joints error, and $\lambda_{\theta/\beta}$ is for the SMPL-X pose and shape parameters $\theta/\beta$. The 2D keypoint loss $\mathcal{L}_{2D}$ consists of the loss of both predicted 2D keypoints and the projected 3D keypoints:

$$\mathcal{L}_{2D} = \mathcal{L}_{img} + \mathcal{L}_{proj}, \quad (8)$$

where $\mathcal{L}_{2D}$ denotes the 2D keypoint prediction error and $\mathcal{L}_{proj}$ denotes the projected 3D keypoint prediction error.

## 4 EXPERIMENTS

### 4.1 Setup

**Datasets.** We train HMR-Adapter for the whole-body model [3] and the hand expert [29] on datasets that provide accurate 3D hand annotations. Specifically, we use Human3.6M [18], RICH [15], and BEDLAM [1]. For training HMR-Adapter for the hand expert, we additionally use FreiHAND [47] and InterHand2.6M [25]. We crop out hand images from whole-body images using the ground-truth hand bounding boxes. For 3D whole-body reconstruction evaluation, we use ARCTIC [7] and EHF [28]. We additionally use 3DPW [36] to evaluate the body-only reconstruction quality. ARCTIC is a hand-object interaction dataset fitted with accurate SMPL-X annotations, containing 10 subjects manipulating 11 objects. Notably, we follow [3] to exclude the egocentric frames in our testing as they only capture hands. The EHF dataset is a widely used dataset to test the performance of expressive HMR, providing SMPL-X aligned 3D mesh that accurately reflects the subject's diverse body, hand, and face articulations. 3DPW is one of the most famous outdoor datasets with 3D SMPL [23] annotations without hand or face annotations.

**Evaluation Metrics.** For whole-body and hand mesh reconstruction, we use Mean Per Vertex Position Error (PVE) and Mean Per Joint Position Error (MPJPE) to evaluate the positions of 3D vertices and 3D joints, respectively. The average 3D joint distance (in $mm$) and mesh vertex distance (in $mm$) are computed between the predicted and ground-truth values, with an initial alignment of the pelvis joint's translation. The pelvis serves as the root joint for the whole body's 3D error computations. For hands, the root joints are the wrists. The Procrustes Aligned [12] Mean Per Joint Position Error (PA-MPJPE) and Procrustes Aligned Mean Per Vertex Position Error (PA-PVE) are calculated by additionally aligning the rotation and scale. The 3D joint coordinates required for the MPJPE and PA-MPJPE calculations are derived by applying a joint regression matrix from the SMPL-X model to the mesh, consistent with prior studies [3, 22, 24]. Average 3D errors for the left and right hands are reported as the 3D hands' error measurement.

**Implementation Details.** In our experiments, we choose SMPLer-X [3] and HaMeR [29] as our baseline whole-body model and hand expert model, respectively. There are 6 attention layers in the HMR-Adapter. The total trainable parameters of our HMR-Adapter include a body pose feature projection network and 6 attention modules, amounting to about 27M in HaMeR and 37M in SMPLer-X. HMR-Adapter is trained on a single machine with 4 V100 GPUs for 10K steps with a batch size of 32 per GPU, which roughly takes 6 hours. The detailed architecture of HMR-Adapter and more training details can be found in the supplementary materials.

### 4.2 Comparison Against Other Methods

**Quantitative Comparison.** Table 1 shows the quantitative results for our proposed method and previous state-of-the-art methods, and we evaluate their performance on whole-body reconstruction and hand reconstruction by PA-PVE, PVE, and PA-MPJPE. We follow [39] to construct a baseline model that combines the pre-trained SMPLer-X-H32 (SMPLer-X [3] with ViT-Huge backbone and trained on 32 different datasets) and HaMeR using Inverse Kinematics [39]. From table 1, we can see that, after adaptation with HMR-Adapter, our pipeline brings significant improvements over the baseline by

| Method | ARCTIC | | | | | | EHF | | | | | |
|---|---|---|---|---|---|---|---|---|---|---|---|---|
| | PA-PVE ↓ | | PVE ↓ | | PA-MPJPE ↓ | | PA-PVE ↓ | | PVE ↓ | | PA-MPJPE ↓ | |
| | All | Hands | All | Hands | All | Hands | All | Hands | All | Hands | All | Hands |
| Hand4Whole [24] | 63.4 | 18.1 | 136.8 | 54.8 | – | – | 50.3 | 10.8 | 76.8 | 39.8 | – | – |
| OSX [22] | 33.0 | 18.8 | 58.4 | 39.4 | – | – | 48.7 | 15.9 | 70.8 | 53.7 | – | – |
| PyMAF-X (HR48) [39] | – | – | – | – | – | – | 50.2 | 10.2 | 64.9 | **29.7** | 52.8 | 10.3 |
| SMPLer-X-B32 [3] | 34.9 | 18.9 | 56.3 | 40.9 | 34.5 | 19.8 | 40.7 | 14.5 | 67.3 | 52.1 | 44.4 | 14.9 |
| SMPLer-X-H32 [3] | 29.3 | 18.9 | 48.5 | 38.3 | 28.9 | 19.2 | 39.0 | 14.8 | **56.9** | 42.2 | 43.2 | 14.7 |
| SMPLer-X-H32 † [3] | 27.7 | 18.8 | 44.7 | 37.0 | – | – | – | – | – | – | – | – |
| RoboSMPLX [27] | – | – | – | – | – | – | 49.7 | **10.0** | 73.7 | 34.9 | – | – |
| SMPLer-X-H32 + HaMeR[29] | 28.3 | 16.4 | 47.0 | 39.0 | 27.8 | 17.4 | 43.0 | 12.3 | 59.7 | 52.2 | 43.5 | 12.1 |
| Ours | **23.9** | **10.5** | **40.7** | **23.6** | **24.0** | **11.0** | **36.8** | **10.0** | 57.1 | 32.4 | **42.0** | **9.9** |

Table 1: Quantitative comparisons against state-of-the-art methods on the EHF [28], ARCTIC [7] dataset. The best performance is in boldface. We use PA-MPVPE (*mm*), MPVPE (*mm*), and MPVPE (*mm*) as evaluation metrics.

13.4% and 39.5% on whole-body and hand PVE on ARCTIC, respectively. Our method also surpasses other competitors in terms of quality for both whole-body and hand reconstruction. On the ARCTIC dataset, our method shows notable improvements in hand reconstruction, showcasing the robustness of our proposed HMR-Adapter in occlusion scenarios. The results underscore the effectiveness of the additional guidance brought by the HMR-Adapter for both the large whole-body model and the hand expert model.

Although our primary focus is on Expressive HMR, we additionally show the quantitative results on 3DPW in Table 2 to demonstrate the improvement of body reconstruction quality and the generalizability of our method. We evaluate the body-only reconstruction performance by PVE, PA-MPJPE, and MPJPE. We can still boost the performance of the baseline model SMPLer-X significantly in terms of body prediction accuracy, affirming that HMR-Adapter enhances body pose estimation in the large whole-body model.

**Qualitative Comparison.** We also present qualitative results in Fig. 3. We can see that large models with our HMR-Adapter can produce more reasonable results for body and hand motion under challenging cases, such as dense hand-object interactions or motion blur. For example, in the (a) instance, our predictions come with more reasonable hand poses for the occluded right hand. In the (c) instance, SMPLer-X gives the erroneous prediction for the person's left elbow, while ours gives a more accurate body pose prediction, proving the efficacy of the proposed dual-path cross augmentation pipeline. In the (f) instance, SMPLer-X produces inaccurate hand poses and upper body poses, while our prediction with refined body pose and hand pose fits the input image better.

To validate the efficacy of our proposed HMR-Adapter on the hand expert model, we further show qualitative results of hand pose predictions in Fig. 4, where we show the input hand images, predicted results by HaMeR with HMR-Adapter, and the original HaMeR's prediction. Compared to the baseline method, our model can give a more reasonable prediction for occluded hands. For example, in the (e) instance, our method gives a reasonable hand global orientation prediction due to the body pose guidance from the arm poses, while the hand-only baseline fails to achieve that. In the (f) instance, our method gives a reasonable finger pose prediction, while the hand-only baseline could only infer from the limited hand image cues, leading to erroneous finger predictions.

| Method | PVE ↓ | PA-MPJPE ↓ | MPJPE ↓ |
|---|---|---|---|
| Body-only (SMPL) Methods | | | |
| POTTER [43] | 87.4 | 44.8 | 75.0 |
| SimHMR [17] | 102.8 | 49.5 | 81.3 |
| HMR 2.0 [11] | **82.2** | **44.4** | **69.8** |
| Whole-Body (SMPL-X) Methods | | | |
| Hand4Whole[24] | – | 54.4 | 86.6 |
| OSX [22] | – | 60.6 | 86.2 |
| PyMAF-X(HR48)[39] | 91.3 | 47.1 | 78.0 |
| RoboSMPLX[27] | 95.2 | 48.5 | 80.1 |
| SMPLer-X-B32[3] | 90.3 | 53.4 | 80.3 |
| SMPLer-X-H32 [3] | 86.1 | 50.6 | 75.0 |
| SMPLer-X-H32 + HaMeR [29] | 88.3 | 46.7 | 78.3 |
| Ours | **82.1** | **44.0** | **73.0** |

Table 2: Evaluation on 3DPW [36]. We compare our HMR-Adapter with previous methods.

## 4.3 Ablation Study

In this section, we first validate our design's ability to fine-tune a target dataset in Table 3 and Table 4 on both hand model and hand whole-body models. Then, we show that the proposed HMR-Adapter is a better way to inject additional body pose guidance into the hand expert model to improve hand prediction by comparing different injection designs. Finally, we validate the design choices in HMR-Adapter for the large whole-body model.

**HMR-Adapter for Fine-tuning** Table 3 demonstrates that using HMR-Adapter to fine-tune the existing large motion models is more efficient and can achieve competitive or superior results than the fully fine-tuned model with much less trainable parameters. First, we show quantitative results of fine-tuning HaMeR on ARCTIC. We crop out the hand images using the ground truth hand bounding boxes as training and testing data. In Table 3, "HaMeR" is the pre-trained HaMeR model without fine-tuning. No.1 is the model that fine-tunes the whole network from the pre-trained model. No.2 trains the whole model from scratch. We train No.2 for double steps of other settings. No.3 only fine-tunes the pre-trained model's decoder head. No.4 follows previous methods [24] to concatenate the original hand image feature and the additional body pose feature as the new input to HaMeR's decoder. It fine-tunes HaMeR's original decoder head and learns an extra body feature projection layer. No.5

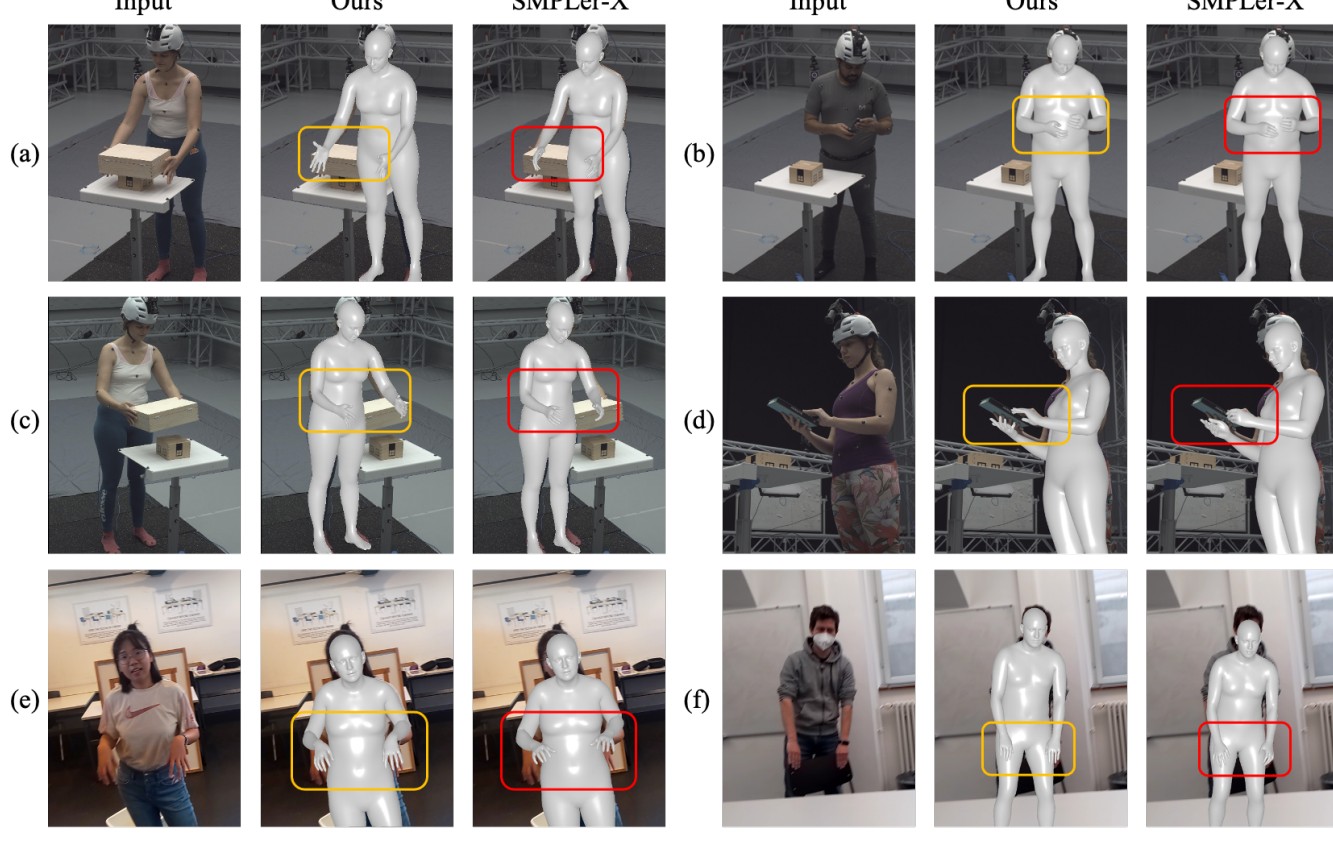

**Figure 3: Qualitative comparison of the baseline model SMPLer-X [3] and ours that use HMR-Adapter to refine the body and hand poses. Using HMR-Adapter significantly improves the 3D hand and upper-body estimation. (a) ~ (d) are from the ARCTIC dataset, and (e) ~ (f) correspond to the EgoBody dataset.**

uses HMR-Adapter, with the hand image feature as HMR-Adapter's input (the original input of HaMeR's decoder head). No.6 is ours, which uses HMR-Adapter to introduce the body pose feature. We can observe that:

- *The pre-trained model helps with the convergence* (No.1 vs. No.2): No.2 takes longer training time but is still worse than No.1.
- *HMR-Adapter is highly efficient* (No.5 vs. No.1, No.3): With no additional input, fine-tuning with HMR-Adapter leads to higher performance even than the fully-trained model. This indicates that HMR-Adapter retains the original model's knowledge and fits new data quickly.
- *The additional body pose information is useful* (No.6 vs. No.5): Both equipped HMR-Adapter, No.6 receives body pose as additional input and thus improves PVE and PA-PVE by 22.0% and 6.7%, respectively.
- *HMR-Adapter is advantageous in injecting body pose information* (No.6 vs. No.4): Concatenating various features [24, 45] can impair hand prediction by introducing irrelevant information. In contrast, our approach preserves the hand expert's original proficiency in hand estimation for typical scenarios and enhances its capability to predict occluded hands

| | Pre-trained | Adapter | Trainable Params. | PA-PVE | PVE | PA-MPJPE |
|---|---|---|---|---|---|---|
| HaMeR | ✓ | – | – | 16.2 | 38.5 | 17.1 |
| No.1 | ✓ | ✗ | 671M | 13.9 | 45.3 | 15.0 |
| No.2 | ✗ | ✗ | 671M | 15.1 | 53.3 | 16.3 |
| No.3 | ✓ | ✗ | 39M | 14.1 | 56.1 | 15.1 |
| No.4 | ✓ | ✗ | 39M | 17.9 | 36.7 | 18.4 |
| No.5 | ✓ | ✓ | 27M | 9.0 | 30.9 | 9.2 |
| No.6 | ✓ | ✓ | 27M | **8.4** | **24.1** | **8.5** |

**Table 3: Hand reconstruction quality on ARCTIC, obtained by different fine-tuning methods on HaMeR [29].**

In Table 4, we also conduct similar experiments for different fine-tuning methods on SMPLer-X. The experiment settings are identical for No.1-No.6. We use hand MCP joints' image features as additional input for No.4 and No.6, while No.5 receives the original body pose task tokens as HMR-Adapter's input.

**Body pose guidance for the Hand Expert.** To verify the advantages of using HMR-Adapter to incorporate body features for predicting 3D hand poses, we further conduct comparison between different injection methods. Table 5 shows the quantitative results. To construct the baselines, we first follow [24] to concatenate hand

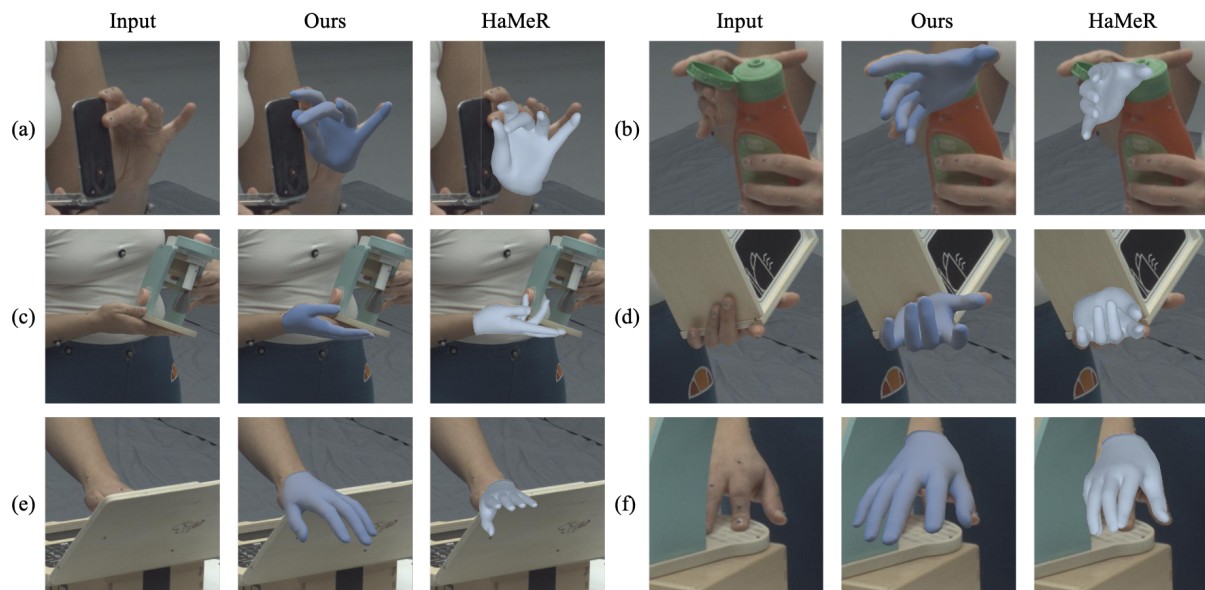

**Figure 4: Qualitative comparison of the baseline model HaMeR [29] and ours that use HMR-Adapter to refine the hand poses. Using HMR-Adapter significantly improves the robustness of 3D hand estimation under occlusion.**

| | Pre-trained | Adapter | Trainable Params. | PA-PVE | PVE | PA-MPJPE |
|---|---|---|---|---|---|---|
| SMPLer-X | ✓ | – | – | 29.3 | 48.5 | 28.9 |
| No.1 | ✓ | ✗ | 662M | 19.5 | **24.4** | 21.1 |
| No.2 | ✗ | ✗ | 662M | 27.0 | 35.0 | 34.6 |
| No.3 | ✓ | ✗ | 31M | 21.0 | 25.1 | 21.2 |
| No.4 | ✓ | ✗ | 31M | 23.6 | 29.9 | 25.1 |
| No.5 | ✓ | ✓ | 37M | 20.8 | 24.7 | 21.2 |
| No.6 | ✓ | ✓ | 37M | **20.7** | 24.7 | **21.0** |

**Table 4: Whole-body reconstruction quality on ARCTIC, obtained by different fine-tuning methods on SMPLer-X [3].**

image features and body pose features, then feed it to the original decoder. We also follow [8] to use a multi-layer perception as a moderator to calculate the confidence for different part features before adding them together. We can see from Table. 5 that HMR-Adapter achieves more accurate 3D hand poses by leveraging the extensive hand estimation training from the pre-trained hand expert and incorporating new body pose guidance. One reasonable explanation is that concatenation harms the model's ability to choose useful information from inputs. Although the weighted summation could avoid some unrelated information using the moderator, it still cannot fully leverage the body conditions. HMR-Adapter disentangles the body pose information and uses the attention mechanism to choose the information that is most related to hand motion while maintaining the original model's strong ability. Besides, as can be seen from the table, it is necessary to discard the lower body poses for body pose guidance. We believe that discarding the lower body poses helps the model to concentrate on the most related information in the upper body.

**Hand guidance for the Whole-body model.** Similarly, we verify the efficacy of using HMR-Adapter on the whole-body model

| Body feature injection methods | PA-PVE (hands) | PVE (hands) |
|---|---|---|
| Concatenation | 15.2 | 41.9 |
| Weighted sum | 12.1 | 26.0 |
| Whole-body pose | 13.4 | 27.7 |
| HMR-Adapter (Ours) | **10.5** | **23.6** |

**Table 5: Ablation of different methods to inject body feature into HaMeR. We report their hand reconstruction errors on ARCTIC.**

| Hand feature injection methods | PA-PVE | PVE | PA-MPJPE |
|---|---|---|---|
| Concatenation | 79.3 | 95.4 | 94.0 |
| Weighted sum | 57.0 | 74.5 | 68.0 |
| HMR-Adapter (Ours) | **36.8** | **57.1** | **42.0** |

**Table 6: Ablation of different methods to inject hand joint feature into SMPLer-X. We report their whole-body reconstruction errors on EHF.**

to incorporate hand features for predicting 3D body poses. The quantitative results are presented in Table 6.

## 5 CONCLUSION

We have proposed HMR-Adapter to boost the performance of existing human mesh recovery models. HMR-Adapter features a dual-path cross augmentation architecture that introduces additional guidance information for body and hand prediction refinement. We seamlessly combine whole-body and hand expert models to validate our approach in expressive human mesh recovery. Extensive experiments prove that HMR-Adapter significantly enhances the robustness and efficiency without requiring extensive fine-tuning, particularly in hand reconstruction quality.

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
