# OpenReview forum: "HMR-Adapter: A Lightweight Adapter with Dual-Path Cross Augmentation for Expressive Human Mesh Recovery"
_acmmm.org/ACMMM/2024/Conference — MM2024 Poster_

### Official Review · Reviewer_5L5y · 2024-05-18

**Rating:** 4
**Confidence:** 3

**Summary:**

This paper proposes a dual-path cross augmentation framework with an HMR-Adapter approach to address the challenges in expressive human mesh recovery from RGB images. The HMR-Adapter enhances the decoding module of large HMR models by injecting additional guidance from other body parts, improving both hand pose and body pose estimations. By combining the adapted large whole-body and hand expert models, HMR-Adapter achieves significant improvements in expressive HMR performance, demonstrating its effectiveness through extensive experiments and analysis.

**Strengths:**

[Novelty] The proposed approach exhibits novelty by leveraging the adapter concept, which effectively builds upon the existing literature on human body and hand pose estimation for expressive human mesh recovery.

[Clarity] This paper is well organized, and descriptions are clear in general.

**Limitations:**

1. It would be beneficial to discuss or clarify certain details of the methods to enhance the strength of the paper's contributions:

1.1 Why is the hand pose derived from the regression of the original decoder, whereas the body pose is predicted through an offset mechanism?

1.2 Why is it that body pose information is utilized to guide the learning of hand pose, whereas it is the hand features, rather than hand pose information, that are employed to direct the learning of body pose?

1.3 Why did the author not consider utilizing face pose information? Given that body pose and hand pose can enhance each other's estimation, how might the inclusion of face pose information contribute?

2. Some points remain unclear:

2.1 What does † refer to in Table 1?

2.2 How is the inference process executed? Are the experimental results for HMR-Adapter in Table 1 achieved by utilizing ground truth (GT) from different branches or solely by directly adopting the baseline model's output? Specifically, are the test results for hands based on reading the GT of body pose or on the prediction results of SMPLer-X-H32? Conducting ablation experiments with these two distinct settings would provide valuable insights.

2.3 “critical information such as the motion of the wrists and arms are getting much greater attention when predicting hand poses.” (lin 152) The paper lacks a corresponding theory or experiment to substantiate this conclusion. If the stated conclusion holds true, the body area influenced by the hand should primarily manifest in the pose expression of the wrist and arm. However, how do we interpret the effect of the hand pose on the overall upper body pose as depicted in Figure 3 (f)?


3. Minor points:

3.1 The representation in Figure 2 can be misleading, as the input for hand pose decoding comprises not only the body pose but also the image features of the hand, leading to a similar issue in the other branch.

3.2 Reference citation format is not uniform.

3.3 There is no discussion on the failure cases or limitations of the introduced method.

**Suitability:**

3

---

### Official Review · Reviewer_QpCV · 2024-05-21

**Rating:** 3
**Confidence:** 2

**Summary:**

This paper presents a dual-path cross-augmentation framework featuring a novel adaptation technique called HMR-Adapter, which enhances the decoding module of large HMR models. The HMR-Adapter significantly boosts the performance of expressive HMR by incorporating guidance from other body parts. This method refines hand pose predictions by integrating body pose information and utilizes additional hand features to improve body pose estimation in whole-body models.

**Strengths:**

1)This paper is easy to understand.

2)The idea of associating and integrating of body pose information  and hand pose information to enhance human mesh recovery is interesting.

3)The proposed method achieves state-of-the-art performance on the ARCTIC and 3DPW datasets.

**Limitations:**

1)The novelty of the approach appears limited. The decoupled transformer module resembles the decoder of the Hand Expert Model. Additionally, the paper employs a two-branch structure and a large pretraining model, but their usage does not seem to differ significantly from other existing methods.To further delve into the idea of associating and integrating body and hand information, the article needs to propose innovative methods.

2)The paper utilizes three models: the Hand Expert Model, the Whole-Body Model, and the proposed HMR-Adapter. Although the HMR-Adapter is parameter-efficient, using three models instead of a single model increases the inference speed and computational resources required during the inference phase.

3)The network aims to improve hand pose estimation, but its performance on the EHF dataset is subpar. The previous method, PyMAF-X (HR48), shows comparable (PVE: 10.2 vs. 10.0, 10.3 vs. 9.9) or even superior (PVE: 29.7 vs. 32.4) results for hand pose estimation.

**Suitability:**

2

---

### Official Review · Reviewer_Eerv · 2024-05-24

**Rating:** 3
**Confidence:** 3

**Summary:**

This paper aims to improve the performance of large-scale Human Mesh Recovery (HMR) models. To this end, the authors propose a dual-path cross augmentation framework, dubbed HMR-Adapter. Specifically, hand features are used to improve body pose estimation while body pose features are adopted to refine hand pose predictions. Experimental results show that the proposed approach considerably improves the performance of baselines.

**Strengths:**

1) This paper presents an efficient approach for finetuning large-scale HMR models.
2) The proposed approach achieves notable improvements over baselines, particularly in hand mesh recovery.

**Limitations:**

1) Insufficient experiments. As this paper aims to develop an efficient adapter for large-scale HMR models, it is necessary to see how the proposed approach performs on foundation models with different capacities. However, the authors only apply their approach to SMPLer-X-H32.
2) The effectiveness of the whole-body adapter seems doubtful. As shown in Table 4, finetuning the decoder head already achieves good performance, where the trainable parameters are less than the proposed adapter.
3) Missing analysis on the effectiveness of the proposed approach. For example, in what scenarios can the proposed approach improve the performance of baseline?
4) The performance gain seems to primarily come from the hand part. Perhaps adding the body pose feature is sufficient to achieve good performance. More justifications are required to validate the effectiveness of the hand adapter.

**Suitability:**

2

---

### Official Review · Reviewer_Xa9r · 2024-05-26

**Rating:** 4
**Confidence:** 4

**Summary:**

This paper introduces a method to estimate the 3D pose and shape of 3D human body and hand from a single image. The idea is to develop a module that integrates SOTA body and hand estimation methods at the feature level to obtain more stable results in occlusion situations. In this way, the hand poses could be inferred based on image feature and upper body poses. The proposed method has been evaluated on ARCTIC and EHF datasets to show the effectiveness. The ablation study shows that, compared with directly integrating the results, such feature-fusion design achieve better peformance on the two benchmarks.

**Strengths:**

1. Promising results. The qualitative results under occlusion cases look really promising, which shows the suprior of the proposed feature-level integration module.
2. The quantitative experiments, especially the ablation study compared with SMPLer-X + HaMeR, validate the effectiveness of the proposed method.
3. The paper is easy to understand.

**Limitations:**

1. Novelty. The proposed method can be a minimal implementation to verify the importance of overall body pose information for more stable hand pose estimation from a whole-body image. However, such idea has been well proved by many previous methods, such as Pymaf-X, Hybrik-X, and SMPLer-X. From this point of view, the technical novelty of the proposed method is quite limited.
2. Overclaim. The proposed method is claimed to be effective due to the small size of the feature fusion module.  A more reasonable way to compare efficiency is to compare the runtime of the entire model.  It is more fair to compare the efficiency of the entire solution rather than just the integrated modules.  Please consider properly accounting for contributions.

**Suitability:**

3

---

### Meta-Review · Area_Chair_8Gut · 2024-07-07

**Recommendation:** Accept (Poster)
**Confidence:** 5

**Metareview:**

All reviewers have agreed to accept this paper. The authors need to address all the reviewers' comments in the revised manuscript.